# Off-Hour Admission Is Associated with Poor Outcome in Patients with Intracerebral Hemorrhage

**DOI:** 10.3390/jcm12010066

**Published:** 2022-12-21

**Authors:** Muhammad Junaid Akram, Xinni Lv, Lan Deng, Zuoqiao Li, Tiannan Yang, Hao Yin, Xiaofang Wu, Mingjun Pu, Chu Chen, Libo Zhao, Qi Li

**Affiliations:** 1Department of Neurology, The First Affiliated Hospital of Chongqing Medical University, Chongqing 400016, China; 2Department of Neurology, Yongchuan Hospital of Chongqing Medical University, Chongqing 402160, China; 3Chongqing Key Laboratory of Cerebrovascular Disease Research, Chongqing 402160, China; 4Department of Neurology, The Second Affiliated Hospital of Anhui Medical University, Hefei 230601, China

**Keywords:** stroke, intracerebral hemorrhage, off-hour, outcome

## Abstract

The mortality of stroke increases on weekends and during off-hour periods. We investigated the effect of off-hour admission on the outcomes of intracerebral hemorrhage (ICH) patients. We retrospectively analyzed a prospective cohort of ICH patients, admitted between January 2017 and December 2019 at the First Affiliated Hospital of Chongqing Medical University. Acute ICH within 72 h after onset with a baseline computed tomography and 3-month follow-up were included in our study. Multivariable logistic regression analysis was performed for calculating the odds ratios (OR) and 95% confidence interval (CI) for the outcome measurements. Of the 656 participants, 318 (48.5%) were admitted during on-hours, whereas 338 (51.5%) were admitted during off-hours. Patients with a poor outcome had a larger median baseline hematoma volume, of 27 mL (interquartile range 11.1–53.2 mL), and a lower median time from onset to imaging, of 2.8 h (interquartile range 1.4–9.6 h). Off-hour admission was significantly associated with a poor functional outcome at 3 months, after adjusting for cofounders (adjusted OR 2.17, 95% CI 1.35–3.47; *p* = 0.001). We found that patients admitted during off-hours had a higher risk of poor functional outcomes at 3 months than those admitted during working hours.

## 1. Introduction

Stroke is the third-leading cause of death and disability, with an incidence of 3 million per year in China [1,2]. Intracerebral hemorrhage (ICH) accounts for 10–30% of all strokes and carries the highest mortality and morbidity among all stroke subtypes [3]. The incidence of stroke increases on weekends, while hospitals lack experienced staff and first-aid providers during this period of time [4,5]. In several studies, a relatively high mortality rate was associated with admissions on weekends and other conditions including cancer, aneurysm and pulmonary embolism [6,7].

There is a controversy whether being admitted to a hospital on “weekends” or during “off-hours” will create adverse consequences to patients [8]. The phenomenon that unfavorable outcomes are correlated with admission time is also called the off-hour or weekend effect, which was raised by researchers as early as the 1970s [9,10]. However, the mechanisms remain unclear. A plausible explanation for the altered clinical outcomes is differences in the patients’ characteristics and the quality of care [11].

During the past few decades, studies have reported that patients admitted to the hospital with acute stroke during off-hours or weekends had a high rate of death and other adverse effects including poor rehabilitative outcomes and delayed medical interventions [4,11,12,13,14,15,16,17]. Comprehensive stroke units and established treatment regimens may overcome the weekend or off-hour effect in ischemic stroke patients [18]; however, mortality was increased 7 and 30 days after off-hour admission for ICH in previous reports [8,12]. Contrary to these findings, the Intensive Blood Pressure Reduction in Acute Cerebral Hemorrhage Trial (INTERACT2) study reported that off-hour admission was not associated with increased mortality in acute ICH [19]. We, therefore, investigated the effect of off-hour admission on in-hospital mortality and 3-month functional outcomes in acute ICH patients.

## 2. Materials and Methods

### 2.1. Characteristics of the Study Population

We retrospectively analyzed our prospective cohort of primary spontaneous ICH patients admitted to the First Affiliated Hospital of Chongqing Medical University from 1 January 2017 to 31 December 2019. The study protocol was approved by the appropriate ethics committee. Written informed consent was obtained from all participants enrolled in our study according to the requirements of our local research ethics board. All protocols in our study were performed in accordance with the Declaration of Helsinki. Inclusion criteria were as follows: (1) age >18 years, (2) spontaneous ICH diagnosed within 72 h of onset, and (3) patients had baseline non-contrast computed tomography (NCCT) images and 3-month follow-up assessment using modified Rankin Scale (mRS). Patients with any of the following conditions were excluded: (1) secondary ICH due to head trauma, brain tumor, cerebral aneurysm, arteriovenous malformation, or hemorrhagic transformation of acute ischemic infarction and (2) primary intraventricular hemorrhage.

All relevant clinical and demographic data were collected from the electronic medical records system, including age, sex, time of symptom onset, risk factors such as hypertension, diabetes mellitus, dyslipidemia, and chronic kidney disease, admission systolic blood pressure (SBP) and diastolic blood pressure (DBP), clinical symptoms, laboratory data, duration of the hospital stay, and complications at discharge. We used the National Institute of Health Stroke Scale (NIHSS) and Glasgow Coma Scale (GCS) to assess stroke severity. Functional outcomes recorded by telephone interviews were evaluated using the mRS after 3 months. Poor functional outcome was defined as 3-month mRS > 3, as previously described [20]. Off-hour admissions were defined as admissions outside the official working hours in China (i.e., 8:00 a.m.–6:00 p.m.).

### 2.2. Image Analysis

NCCT scans were performed by using standard clinical parameters with axial 5 mm slice thickness. Radiological variables including location and volume of hematoma were assessed by using computed tomography (CT) scan images. Furthermore, the semi-automated computer-assisted volumetric analysis (AnalyzeDirect medical imaging software, version 11.0; AnalyzeDirect, Inc., Overland Park, KS, USA) was used for measuring hematoma volumes and IVH volumes. A semi-automated threshold-based approach (range of 44–100 Hounsfield Units) was applied to identify hematoma regions, as previously described [20]. Hematoma and ventricular hemorrhage were manually traced in the hyperdense area surrounding the hematoma on each slice and calculated with AnalyzeDirect software. All NCCT scans were independently evaluated by two experienced neurologists. Both readers were blinded to all clinical information. 

### 2.3. Statistical Analysis

Statistical analysis was performed by using IBM SPSS software (Version 26. IBM Corporation, Armonk, NY, USA). Demographic data and baseline characteristics on admission were summarized as mean ± standard deviation (SD), median or interquartile range (IQR) for continuous variables, and number (%) for discrete variables. The total study population was classified into on-hour admission and off-hour admission in terms of time of admission. The differences between the two groups were determined using the Student’s t-test or the Mann–Whitney U test for continuous variables and the χ^2^ or Fisher exact tests for categorical variables, as appropriate. The association between off-hour admissions with functional outcomes was estimated using logistic regression models. We performed multivariate logistic regression analyses using the forward stepwise method, with removal based on likelihood ratio. We also performed an additional multivariable logistic regression analysis defining poor outcome after ICH as mRS > 3 after three months. We also evaluated tolerance and variance inflation factor (VIF) values to assess multicollinearity between variables. *p* value less than 0.05 was considered statistically significant in our analyses.

## 3. Results

### 3.1. Baseline Characteristics

During the study period, a total of 656 patients (453 males and 203 females) with primary ICH met our inclusion criteria. Of those, 338 patients (51.5%) were admitted during off-hours. In the on-hour admission group, the mean age was 61.7 years, the median NIHSS score was 8.0 (IQR: 3.0–16.0), and the median baseline hematoma volume was 11.3 mL (IQR: 4.0–26.9 mL). The average time of NCCT assessment from stroke onset was 6.5 h (IQR: 2.4–23.3 h). In the off-hour admission group, the mean age was 59.5 years, the median NIHSS score was 9.0 (IQR: 4.0–18.0), and the median baseline hematoma volume was 10.6 mL (IQR: 5.0–25.8 mL). The average time from onset to CT was 4.0 h (IQR: 1.7–13.8 h). Patients with an off-hour admission had a shorter time from onset to imaging and higher systolic blood pressure at admission (Table 1). The clinical characteristics between those with and without a poor functional outcome are illustrated in Table 2. Patients with a poor outcome were more likely to have lower admission GCS scores, 8 (IQR 6–13) vs. 15 (IQR 14–15); higher NIHSS scores, 23 (IQR 13–37) vs. 5 (IQR 2–10); a larger hematoma volume at baseline; and a shorter time from symptom onset to baseline CT scan and were more likely to be admitted during off-hours. 

### 3.2. Off-Hour Admission and Outcome in ICH Patients

The distribution of mRS scores in patients during on-hours and off-hours is shown in Figure 1. At the 3-month follow-up, 450 (68.6%) of the 656 ICH patients were functionally independent (mRS score, 0–3). Univariate regression analysis revealed that age (*p* = 0.002), admission systolic blood pressure (*p* = 0.043), the presence of intraventricular hemorrhage (*p* < 0.001), admission GCS score (*p* < 0.001), admission NIHSS score (*p* < 0.001), baseline hematoma volume (*p* < 0.001), time from onset to CT (*p* < 0.001), and off-hour admission (*p* = 0.008) were associated with a poor functional outcome (Table 3). After adjusting for age, presence of intraventricular hemorrhage, admission systolic blood pressure, admission GCS score, and time from onset to CT, off-hour admission remained as an independent predictor of the outcome (OR 2.17, 95% CI 1.35–3.47; *p* = 0.001). We also evaluated the tolerance and VIF values to assess the multicollinearity between the variables, as shown in Appendix A.

## 4. Discussion

We found that off-hour admission occurs in approximately half of patients with ICH and is an independent predictor of a 3-month poor outcome. Our findings suggest that it is imperative to improve the quality of care for ICH patients during off-hours. In recent years, several studies have investigated the “off-hour effect” in acute stroke patients, finding that patients admitted during an off-hour period had an unfavorable outcome [4,5,8,14,15,17]. This effect may be due to a reduced quality of care or more severe neurological deficit at presentation. A Dutch study of 82,219 ischemic stroke patients suggested increased death incidence from midnight until 7 AM and decreased death odds from 14:00 until 18:00 [5]. Another study found that the “weekend effect” on mortality appeared to be a decreasing trend over time after a stroke [14]. An analysis of population-based data from 187669 acute ischemic stroke and 34,845 acute hemorrhagic stroke admissions in the GWTG-Stroke program found that presentation during off-hours was significantly associated with higher in-hospital mortality for both ischemic and hemorrhagic stroke admissions [8]. However, these studies were retrospective analysis without detailed information regarding the severity of stroke on admission. A German study of 37,396 patients demonstrated that patients’ admissions on weekends led to increased mortality and complications, though this was no longer significant after adjusting for the clinical state on presentation [21]. In contrast, studies from Canada and the United States have shown no association between weekend admission and stroke mortality [6,18,22,23]. One possible explanation is patients who are admitted to comprehensive stroke centers, where the quality of medical resources might be excellent during off-hours as well as on-hours. This off-hour effect was found to be higher for hospitals in rural areas than for stroke centers or university hospitals [4]. Of note, only a few prior studies were adjusted for the severity of stroke, which is one of the strongest predicting factors for outcomes [24]. 

To the best of our knowledge, this is the first large sample study evaluating the off-hour effect on the functional outcome in Chinese patients with ICH. Our results suggest that patients admitted during off-hours had poor outcome after 3 months, after adjusting for GCS score on presentation. Since most hospitals were short of staff during off-hours and China lacks a strengthened organization system for acute ischemic stroke, it is imperative to establish a streamlined management protocol for ICH patients. Few studies investigated the “off-hour/weekend effect” with regard to intracerebral hemorrhage conditions. A retrospective cohort study including 13821 patients showed a correlation between off-hour hospital admission and higher mortality [12]. However, some studies have shown no off-hour effect on ICH patients [9,19,25]. A recent registry study of 1269 ICH subjects detected an association between off-hour hospital presentation and functional outcomes rather than short-term mortality [9]. In this study, 7-day mortality was not significantly higher after adjusting for confounding variables. Furthermore, the INTERACT2 study of 2794 subjects reported that off-hour admission was not associated with an increased risk of mortality or morbidity in acute ICH patients [19]. With regard to strict blood pressure control treatment, patients showed a significant reduction in systolic blood pressure levels, indicating that established management standards can result in a similar therapeutic effect irrespective of admission hours [19]. In a post hoc analysis of the Anti-hypertensive Treatment of Acute Cerebral Hemorrhage (ATACH-2), subjects who received nicardipine within 2 h of onset had a lower frequency of ICH expansion in the intensive blood pressure reduction group and an improved outcome [26]. Therefore, in some medical hospitals, it seems necessary that ultra-early and standardized treatment strategies and the implementation of standard procedures may overcome the “off-hour effect”, at least in noninvasive managements after ICH. Efforts aimed at decreasing the disability and mortality rate among hemorrhagic stroke patients at the early stage seem warranted. 

Off-hour admissions have been shown to have a negative effect on clinical outcomes in our study, demonstrated by the longer lengths of hospital stays, likely due to factors related to limited off-hours staffing. The discrepancy with our study may be due to the difference in the overall proportion of patients admitted or the availability of resources during off-hours. Another Canadian study revealed that patients admitted during the weekend had a higher rate of mortality and a lower likelihood to of being discharged to their residence. There was a 14% increase in the risk of mortality, after adjusting for age, gender, comorbidities, and certain other major medical conditions [4].

Our study also had some limitations. Firstly, our study was based on a single cohort with a limited sample size. Secondly, the differences in the amount and expertise of the hospital staff and the staff present at the time of admission were not measured in this study. Future studies with a larger sample size and involving measurements of the staff present at the time of admission should be conducted to further clarify the effect of off-hour admission on functional outcome in ICH patients.

## 5. Conclusions

Our study concluded that patients admitted during off-hours had a higher risk of poor functional outcome in the long term than those admitted during working hours. There is an increasing need to establish healthcare programs to improve care for ICH patients during off-hours.

## Figures and Tables

**Figure 1 jcm-12-00066-f001:**
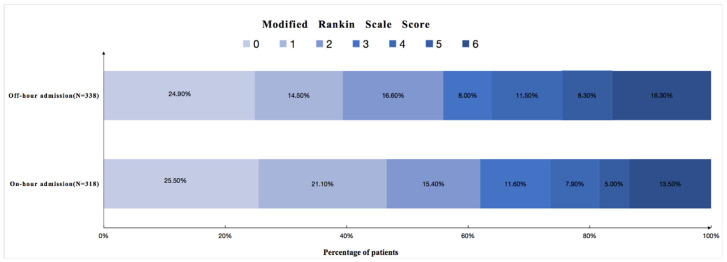
Distribution of modified Rankin Scale score in patients with on-hour admission and off-hour admission.

**Table 1 jcm-12-00066-t001:** Demographic data and baseline characteristics of ICH patients at time of admission (*n* = 656).

Variable	On-Hour Admission *n* = 318 (48.5%)	Off-Hour Admission *n* = 338 (51.5%)	*p* Value
** *Demographics* **			
Mean age, years (SD)	61.7 (14.2)	59.5 (14.0)	0.540
Sex, male, *n* (%)	225 (70.8%)	228 (67.5%)	0.361
** *Medical history* **			
Smoking, *n* (%)	150 (47.2%)	145 (42.9%)	0.316
Alcohol consumption, *n* (%)	114 (35.8%)	98 (29.0%)	0.094
History of hypertension, *n* (%)	227 (71.4%)	240 (71.0%)	0.761
History of diabetes mellitus, *n* (%)	54 (17.0%)	55 (16.3%)	0.802
History of ischemic stroke, *n* (%)	36 (11.3%)	28 (8.3%)	0.211
History of hemorrhagic stroke, *n* (%)	20 (6.3%)	23 (6.8%)	0.745
Previous antiplatelet use, *n* (%)	13 (4.1%)	16 (4.7%)	0.688
Previous antihypertensive use, *n* (%)	111 (34.9%)	111 (32.8%)	0.576
Statin use before events, *n* (%)	15 (4.7%)	13 (3.8%)	0.549
** *Clinical features* **			
Median SBP, mmHg (IQR)	168.0 (151.0–186.0)	177.0 (154.5–198.5)	0.004
Median DBP, mmHg (IQR)	97.0 (84.0–108.0)	99.0 (84.0–113.0)	0.057
Median admission GCS score (IQR)	14.0 (12.0–15.0)	14.0 (10.0–15.0)	0.066
Median admission NIHSS score (IQR)	8.0 (3.0–16.0)	9.0 (4.0–18.0)	0.210
Median time from onset to imaging, h (IQR)	6.5 (2.4–23.3)	4.0 (1.7–13.8)	<0.001
Median hematoma volume, ml (IQR)	11.3 (4.0–26.9)	10.6 (5.0–25.8)	0.683
IVH at baseline CT, *n* (%)	108 (34.0%)	123 (36.4%)	0.515
Pulmonary infection, *n* (%)	110(34.6%)	132(39.1%)	0.173
Urinary tract infection, *n* (%)	25(7.9%)	24(7.1%)	0.720

ICH: intracerebral hemorrhage, GCS: Glasgow Coma Scale, IVH: intraventricular hemorrhage, IQR: interquartile range, SD: standard deviation, CT: computed tomography, NIHSS: National Institute of Health Stroke Scale, SBP: systolic blood pressure, DBP: diastolic blood pressure.

**Table 2 jcm-12-00066-t002:** Comparison of baseline demographic, clinical, and radiological characteristics between patients with and without poor outcome.

Variables	Patients, No. (%)		
	Good Outcome (*n* = 450, 68.6%)	Poor Outcome (*n* = 206, 31.4%)	*p* Value
**Demographics**			
Mean age, y (SD)	59.4 (13.9)	63.1 (14.5)	0.653
Sex, male, *n* (%)	321 (71.3)	132 (64.1)	0.062
**Medical history**			
Hypertension, *n* (%)	322 (71.6)	145 (70.4)	0.879
Diabetes mellitus, *n* (%)	71 (15.8)	38 (18.4)	0.307
Prior ICH, *n* (%)	21 (4.7)	22 (10.7)	0.003
Prior ischemic stroke, *n* (%)	39 (8.7)	25 (12.1)	0.139
**Clinical features**			
Systolic blood pressure, mmHg (SD)	171.0 (28.9)	176.2 (32.6)	0.053
Diastolic blood pressure, mmHg (SD)	99.5 (18.1)	96.5 (21.3)	0.068
Median admission GCS score (IQR)	15 [14–15]	8 [6–13]	<0.001
Median admission NIHSS score (IQR)	5 [2–10]	23 [13–37]	<0.001
Baseline ICH volume, mL (IQR)	8.5 [3.3–16.7]	27.0 [11.1–53.2]	<0.001
Time from onset to CT, h (IQR)	6.5 [2.5–22.3]	2.8 [1.4–9.6]	<0.001
Pulmonary infection, *n* (%)	111(24.7)	131(63.6)	<0.001
Urinary tract infection, *n* (%)	29(6.4)	20(9.7)	0.111
Off-hour, *n* (%)	216 (48)	122 (59.2)	0.008
**Outcome**			
30-day mortality, *n* (%)	0 (0)	83 (40.3)	<0.001
90-day mortality, *n* (%)	0 (0)	98 (47.6)	<0.001
Median 90-day mRS score (IQR)	1 [0–2]	5 [4–6]	<0.001

ICH: intracerebral hemorrhage, GCS: Glasgow Coma Scale, IVH: intraventricular hemorrhage, IQR: interquartile range, SD: standard deviation, CT: computed tomography, NIHSS: National Institute of Health Stroke Scale, SBP: systolic blood pressure, DBP: diastolic blood pressure, mRS: modified Rankin scale.

**Table 3 jcm-12-00066-t003:** Univariate and multivariate analysis of predictors for poor outcome at 3 months.

Variable	Odds Ratio	95% Confidence Interval	*p* Value
**Univariate analysis**			
Age, year ^a^	1.02	1.01–1.03	0.002
Sex, male	0.72	0.51–1.02	0.063
History of hypertension	1.03	0.71–1.49	0.879
History of diabetes mellitus	1.25	0.81–1.94	0.307
SBP, mmHg ^a^	1.01	1.00–1.01	0.043
DBP, mmHg ^a^	0.99	0.98–1.00	0.069
Admission GCS score ^a^	0.65	0.61–0.70	<0.001
Admission NIHSS score ^a^	1.15	1.13–1.18	<0.001
Baseline ICH volume, mL ^a^	1.05	1.04–1.06	<0.001
IVH at baseline CT	5.72	4.00–8.18	<0.001
Time from onset to CT, hour	0.97	0.96–0.98	<0.001
Off-hour admission	1.57	1.13–2.20	0.008
**Multivariate analysis**			
Age, year	1.05	1.03–1.07	<0.001
Admission GCS score ^a^	0.70	0.65–0.75	<0.001
Baseline ICH volume, mL ^a^	1.28	1.11–1.47	0.001
IVH at baseline CT	2.00	1.24–3.22	0.005
Off-hour admission	2.17	1.35–3.47	0.001

ICH: intracerebral hemorrhage, GCS: Glasgow Coma Scale, IVH: intraventricular hemorrhage, CT: computed tomography, NIHSS: National Institute of Health Stroke Scale, SBP: systolic blood pressure, DBP: diastolic blood pressure; mRS: modified Rankin scale; IQR: interquartile range. ^a^ Per unit change in regressor.

## Data Availability

The datasets generated for this study are available on request to the corresponding author, Q.L. (qili_md@126.com).

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
