# Peer review of "Off-Hour Admission Is Associated with Poor Outcome in Patients with Intracerebral Hemorrhage"

_jcm, 2022, doi:10.3390/jcm12010066_

Round 1

Reviewer 1 Report

In this manuscript Akram et al. studied the effect of off-hour admission in patients with ICH. Authors found that patients admitted at off-hours had a two-fold increased risk of poor outcome compared to patients admitted at on-hours. These results come from a well-phenotyped uni-centric cohort of Chinese patients.

The manuscript results interesting and covers a clinically relevant topic. Authors correctly addressed most confounding factors and draw a clear conclusion. I have several concerns though:

- Authors constructed a multivariate model in table 3 and correctly assessed multicollinearity by the calculation of tolerance and vif (table 4). If my interpretation is correct, authors introduced in the model those variables having a p-value < 0.1 in the univariate analysis (methods section: ”in the logistic regression models, all variables with p<0.1 were included”). I would recommend changing this approach and constructing the model using a forward stepwise selection method; giving the opportunity to enter into the model other covariables that might result informative.

- According to figure 1, it looks like differences are especially related to mRS-6, such that patients admitted at off-hours had increased mortality. Could you please provide data on the causes of death? (infections, increase in size of the bleeding, ICH recurrence…).

- Authors acknowledge: “ staff present at the time of admission was not measured in this study”, but is there any possibility to recover this data? This might provide important insight.

Minor comments:

- Do authors think that might be interesting to stratify off-hours according to the admission time/day? (weekend admission, vacations…).

- I would suggest moving table 4 to the supplemental material.

Many thanks for interesting work.

Reviewer 2 Report

Manuscript ID: jcm-2053501

Title: ATRIAL Off-Hour Admission is Associated with Poor Outcome in Patients with Intracerebral Hemorrhage

Authors: Muhammad Junaid Akram et. al.

Journal: Journal of Clinical Medicine

This paper is an interesting piece of work and overall well-written. The following comments refer to the statistical analysis of the paper.

Statistical Review

The authors perform a multivariate analysis on poor outcome at 3 months, with covariates selected as the result of a corresponding univariate analysis. From a statistical perspective, it is recommended to start with a model where all covariates of interest are considered (e.g. Table 1) and after proper stepwise regression methods end up with a model where only significant associations are included. The reason for implementing such a strategy is to ensure that the significance of a predictor with the response is assessed when other predictors are present.

Also, a general comment is that it might be more appropriate to assess the effect of off-hour hospital admission on an outcome that is closer in time than the Rankin score after 3 months. For example, one could analyze death occurrence, or length of hospital stay, or clinical condition at hospital discharge. The authors mention such studies in the “Discussion” section and it might be interesting to explore if the results found in these studies are supported by the current study.

Date:    7 December 2022

Round 2

Reviewer 1 Report

Authors conducted a good revision of the manuscript. Thank you for all the answer.